



# Applying prior correlations for ensemble-based spatial localization

Chu-Chun Chang, Eugenia Kalnay

Department of Atmospheric and Oceanic Science, University of Maryland, College Park, United States

*Correspondence to*: Chu-Chun Chang (cchang75@umd.edu)

**Abstract.** Localization is an essential technique for ensemble-based data assimilations (DA) to reduce the sampling errors due to limited ensembles. Unlike traditional distance-dependent localization, the *correlation cutoff method* (Yoshida and Kalnay, 2018; Yoshida 2019) tends to localize the observation impacts based on their background error correlations. This method was initially proposed as a variable localization strategy for coupled systems, but it also can be extensively utilized as a spatial localization. This study introduced and examined the feasibility of the correlation cutoff method as an alternative

spatial localization preliminary on the Lorenz (1996) model. We compared the accuracy of the distance-dependent and correlation-dependent localizations and extensively explored the potential of integrative localization strategies. Our results suggest that the correlation cutoff method can deliver comparable analysis to the traditional localization more efficiently and with a faster spin-up. These benefits would become even more pronounced under a more complicated model, especially when the ensemble and observation sizes are reduced.

## 15 1 Introduction

The ensemble Kalman filter (EnKF) is widely employed in modern numerical weather prediction (NWP) for refining model initial conditions and improving forecasts (Evensen 2003). One of the notable features of EnKF is its flow-dependent background error covariance derived from the background ensembles (e.g., forecasts initialized at the last analysis time), which involves the time-evolving error statistics for the model state. The implied background error covariance and the

observation error covariance together determine how much observation information should be used to generate a new analysis. Therefore, the accuracy of the background error covariance estimates is one of the most critical keys toward an optimal analysis for EnKF.

Houtekamer and Mitchell (1998) noticed that the background error covariance estimated by too few ensembles would introduce spurious error correlations in the assimilation. Incorrect error correlations are harmful to the analysis and could

lead to a filter divergence. Hamill et al. (2001) performed conceptual experiments demonstrating how existing noises in the background error covariance influence the EnKF analysis. Their results showed that the relative error, also known as the noise-to-signal ratio, significantly increases when the ensemble size is reduced, and a large relative error would consequently degrade the analysis accuracy. These early studies concluded that a sufficient ensemble size is essential for EnKF to obtain reliable background error estimates and generate accurate analysis. However, having large ensembles is computationally



expensive, especially for high-resolution models. Hence, finding a balance between accuracy and computational cost becomes an inevitable challenge for modern EnKF applications. Recent EnKF studies usually limit their ensemble size to about 100 members, and the ensemble size employed in operational NWPs is even less due to the consideration of computational efficiency (Houtekamer and Zhang, 2016; Kondo and Miyoshi, 2016).

In order to reduce the sampling errors induced by limited ensembles, covariance localization has become an essential
technique for EnKF applications. Traditionally, localization tends to limit the effects from distant observations (Houtekamer and Mitchell 1998; Hamill et al., 2001), and a straightforward way to implement that is to apply a Schur product, where each element in the ensemble-based error covariance is multiplied by an element from a prescribed correlation function (Houtekamer and Mitchell (2001)). The most widely used prescribed correlation function is the Gaussian-like, distance-dependent function proposed by Gaspari and Cohn 1999 (hereafter GC99). The GC99 function generally assumes that the
observations farther from the analysis grid are less correlated (and even uncorrelated beyond a finite distance); as a result, the impact from distant observations would be suppressed on the analysis during assimilation.

However, the employment of distance-dependent localization also brings several issues and concerns, such as losing distant information and producing unbalanced analysis (Miyoshi et al., 2014; Mitchell et al., 2002; Lorenc, 2003; Kepert 2009). By utilizing a 10240-member EnKF to investigate the true error correlations of atmospheric variables, Miyoshi et al.
(2014) found that continental-scale, even planetary-scales, error correlations certainly exist in atmospheric variables. Thus, the use of distance-dependent localization would artificially remove the real long-range signals from the analysis increments. Another follow-up experiment with the 10240-member EnKF showed that the removal of localization could significantly improve the analysis and its subsequent 7-day forecasts, and the key component for these improvements is the long-range correlation between distant locations (Kondo et al. (2016)).

The imbalance analysis is another noteworthy issue for localization (Cohn et al., (1998); Lorenc (2003); Kepert (2009)). An excellent paper from Greybush et al. (2011) summarized the unbalanced problem induced by localization. They argued that the imbalance analysis could happen for either B or R localizations, and the EnKF analysis accuracy could be affected by the manually defined localization length in GC99. The B and R localizations indicate whether the localization function is applied on the background error covariance B or the observation error covariance R. Furthermore, they found that the B
localization has a longer optimal localization length with respect to the analysis accuracy. In contrast, the R localization is more balanced than the B localization underlying the same localization length, and the balance of the analysis is enhanced when the localization length increases. A similar conclusion is mentioned in Lorenc (2003) that the unbalance induced by localization would relax with longer localization length and significantly minimized when the length is larger than 3000 km.

This study introduces a novel non-adaptive, correlation-dependent localization scheme evolved from the correlation cutoff method (Yoshida and Kalnay, 2018; hereafter, YK18). The key idea is to "localize" the information from observation to analysis according to their square background error correlations estimated from a preceding offline run. Although YK18 was proposed initially as a variable localization strategy for coupled systems, it can be further utilized as a spatial localization



through appropriate employment of the cutoff function (Yoshida 2019). This paper investigates the feasibility and
characteristics of these two types of spatial localization methods, the traditional distance-dependent and the correlation-dependent, on the EnKF applications. Furthermore, we explored the potential of the hybrid use of GC99 and YK18 under different configurations, aiming to gain more insights into integrative localization applications. Note that this study primarily focuses on the impact of non-adaptive localization, so the discussion of adaptive localization (e.g., such as ECO-RAP, Bishop and Hodyss, 2009) is beyond the scope of this paper.

This paper is organized as follows. Section 2 provides brief introductions for the data assimilation (DA) and localization methods. Section 3 describes the model and experiment configurations employed in this study. The results of these experiments are presented in Section 4. Finally, section 5 concludes our findings and future applications.

## 2 Methodology

### 2.1 The local ensemble transform Kalman filter (LETKF)

The LETKF (Hunt et al., 2007) is one of the most popular ensemble-based DA schemes. Its analysis is derived independently at each model grid by combining the local information from the ensemble backgrounds and the observations. At each analysis time, the analysis equations are expressed as:

$$\overline{x_a} = \overline{x_b} + \mathbf{X}_b \, \widetilde{\mathbf{P}_a} \, (\mathbf{HX}_b)^{\mathrm{T}} \mathbf{R}^{-1} [\boldsymbol{y}_o - \mathbf{H}\overline{x_b}] \, , \tag{1}$$

$$\boldsymbol{X}_a = \boldsymbol{X}_b \left[ (k-1) \, \widetilde{\mathbf{P}_a} \right]^{\frac{1}{2}} \, , \tag{2}$$

$$\widetilde{\mathbf{P}_a} = \left[ (k-1)\mathbf{I}_{k \times k} + (\mathbf{H}\boldsymbol{X}_b)^T \mathbf{R}^{-1}(\mathbf{H}\boldsymbol{X}_b) \right]^{-1} \, , \tag{3}$$

where subscript letters $a$ and $b$ denote the analysis and background, respectively. The $\boldsymbol{X}_{(.)}$ represents the matrix of ensemble perturbations where each column is the vector of the deviations from the mean state $\overline{x_{(.)}}$, namely $\boldsymbol{X}_{(.)} = \{(\boldsymbol{x}_{(.)}^i - \overline{x_{(.)}}) | ... | (\boldsymbol{x}_{(.)}^k - \overline{x_{(.)}})\}$ and $\boldsymbol{x}_{(.)}^i$ is the state vector of the $ith$ ensemble with an ensemble size $k$. $\mathbf{H}$ is the observation operator that converts information from model space to observation space. $\boldsymbol{y}_o$ denotes the local observations, and $\mathbf{R}$ is the corresponding observation error covariance. $\widetilde{\mathbf{P}_a}$ is the analysis error covariance in a k-dimensional ensemble space spanned by the local ensembles. This attribute avoids the direct calculation of the error covariance in the M-dimensional model space (given that usually M >> k in NWP applications), and thus, the analysis can be obtained in a very efficient manner.

Since the background error covariance $\mathbf{P}_b$ in LETKF is derived in spanned ensemble space, it is impossible to implement the localization function directly on the background error covariance through the Schur product in physical space like Hamil et al. (2001). Instead, Hunt et al. (2007) proposed another brilliant way to implement localization for LETKF by simply multiplying the elements of $\mathbf{R}^{-1}$ by an appropriate localization weight range from zero to one. This feature, where the





localization function works at the R matrix, is also known as the R localization. The characteristics of R localization and its differences to B localization were discussed in Greybush et al. (2011).

## 2.2 Distance-dependent localization

Following Hunt et al. (2007), we use the positive exponential function as the localization function:

$$\rho_{i_j} = exp\left[\frac{d(i,j)^2}{2L^2}\right],$$
(4)

where $\rho_{i_j}$ is the localization weight and $d(i,j)$ is the distance between the *ith* analysis grid and the *jth* observation. L is the localization length which is usually manually defined. Equation (4) is a smooth and static Gaussian-like function that offers the same localization effect as the GC99 when applied to LETKF. Since the observation errors are assumed to be 100   uncorrelated in our experiments (R is diagonal), the localization weight would be independently assigned for the assimilated observation *j* and analysis grid *i*. So, when the distance ($d(i,j)$ in eq(4)) increases, a larger value of $\rho_{i_j}$ would be multiplied to R, assuming a larger observation error for the jth observation. That would lead to a smaller value in the rows of the Kalman gain ($X_b[(k-1)I + (HX_b)^T R^{-1}(HX_b)]^{-1}(HX_b)^T R^{-1}$) of LETKF, resulting in a smaller weighting for the observation on updating the background; thus, the impact of distant observations would be suppressed on the analysis. When 105   the compact support is presented with the localization function, the observations located beyond a certain distance (in this study is 3.65 times L) from the analysis grid would be discarded by assuming $\rho_{i_j} = 0$.

## 2.3 The correlation cutoff method

The correlation cutoff method (Yoshida and Kalnay, 2018; Yoshida, 2019), a pioneering localization approach for coupled systems, localizes the information from observation to analysis according to their ***square background error correlations***. 110   This method is carried out with two steps:

**Step 1. Obtaining the square error correlation from an offline run**

The prior square error correlations are collected from a preceding offline run. At each analysis time t, an instantaneous background ensemble correlation between the *ith* analysis grid and the *jth* observation is computed as:

$$corr_{i_j}(t) = \frac{\sum_{k=1}^{K}[x_{k_i}(t) - \overline{x_i(t)}][h_j(x_k(t)) - \overline{h_j(x_k(t))}]}{\sqrt{\sum_{k=1}^{K}[x_{k_i}(t) - \overline{x_i(t)}]^2}\sqrt{\sum_{k=1}^{K}[h_j(x_k(t)) - \overline{h_j(x_k(t))}]^2}} ,$$
(5)

where $x_{k_i}(t)$ is the state vector of the *kth* ensemble at the ith analysis grid at time *t*. $h_j(x_k(t))$ is the interpolation to the background state $x_k(t)$ from the analysis grid to the *jth* observation location. The symbol $\overline{(\ )}$ denotes the ensemble mean of a given vector. K is the total ensemble size.

Then, the temporal mean of the squared correlation is computed by:




$$< corr_{i_j}^2 >= \frac{1}{T} \sum_{t=1}^{T} corr_{i_j}^2(t),$$ (6)

T is the total analysis cycles in the offline run. In Yoshida and Kalnay (2018), this prior error correlation is used as a criterion for the variable localization in the coupled DA, where only the correlated observations would be assimilated. For the spatial localization approach, the value, $corr_{i_j}^2$ will serve as the "prior error correlation" to estimate the localization function as described in Step 2.

**Step 2. Converting the prior error correlation into the localization weighting**

The localization function is derived by substituting the prior error correlation obtained in Step 1 to a chosen cutoff function. Here, we followed Yoshida (2019) using the quadratic function as our choice of the cutoff function. The localization weight $\rho_{i_j}$ assigned for the jth observation at the ith analysis grid can be written as:

$$\rho_{i_j} = \begin{cases} 0 & (x \leq c), \\ 1 - \left(\frac{1-x}{1-c}\right)^2 & (c < x \leq 1), \\ 1 & (x > 1) \end{cases}$$ (7)

Where $x = < corr_{i_j}^2 >$ and $c$ is a tunable parameter that defines the slope for the function. We set $c$ equal to 0.05 and 0.01 for the classic and the variant L96 experiments, respectively. The primary purpose of using the cutoff function is to generally smooth out small perturbations and ensure the weight range is between 0 and 1.

An additional threshold is applied to exclude observations with a square error correlation smaller than 1/(K-1). This threshold is chosen because the squared sample correlation estimated by K random samples extracted from an uncorrelated

distribution would converge to 1/(K-1) (Pitman, 1937). So, any value not much larger than 1/(K-1) is assumed to be unreliable (Yoshida, 2019).

**3. Experimental Design**

We carried out a series of experiments with LETKF on the classic and variant Lorenz (1996) models to investigate the fundamental characteristics of the two types of localizations and explore the feasibility of integrative localizations

**3.1 The classic and variant Lorenz models**

The classic Lorenz model (hereafter L96 model; Lorenz and Emanuel, 1998) is a one-dimensional, univariate simplified atmospheric model that consists of a nonlinear term (e.g., representing advection), a linear term (e.g., representing mechanical or thermal dissipation), and an external forcing. The governing equations are:

$$\frac{dX_i}{dt} = (X_{i+1} - X_{i-2})X_{i-1} - X_i + F \, (+f_i),$$ (8)





where the model variable is denoted by $X_i$, $i = 1, ..., M$, and $M = 40$. F is the constant external forcing and is set to be 8 here. The variables form a cyclic chain, where $X_{-1} = X_{M-1}$ and $X_0 = X_M$. The varying forcing term $f_i$ is neglected for the classic L96 model. The model is integrated with the fourth-order Runge-Kutta scheme with a time step of 0.0125 units (four steps correspond to 6 hours). The model was initialized by adding a single random perturbation onto the rest state and integrating for 90 days to remove the model spin-up.

A variant L96 model with a spatially varying forcing $f_i$ appending to the L96 model is used to mimic a more sophisticated model dynamic. We constrained the total external forcing $(F + f_i)$ with a value range of 6 to 10, ensuring the model dynamic remains chaotic and has a wavenumber of 8. This additional forcing characterizes a land-ocean pattern (Figure 1 (a)), where the land region has an irregular and larger forcing (e.g., source), and the ocean region has a uniform and smaller forcing (e.g., sink). As discussed in Lorenz and Emanuel (1998), the primary influence of the changes in F is on its error growth rate.

They found that increasing F has only a little effect on the qualitative appearance of the wave curves, while the error doubling time has an observable decrease.

To understand the fundamental properties of the variant L96 model, we examined the bred vectors (BV, Toth and Kalnay, 1993, 1997, Kalnay, 2002) of the two models. BV is a nonlinear generalization of the leading Lyapunov vectors (see Toth and Kalnay, 1993 and 1997 for a more detailed exposition). Their growth rate is calculated as $\frac{1}{n\Delta t} \ln (\|\delta x^f\| / \|\delta x^0\|)$, where

$\delta x^f$ and $\delta x^0$ are the final and initial perturbations within the breeding window, respectively. $n$ is the window size and $\Delta t$ is the integration step. The growth rate can be seen as a measure of the local instability of the flow. Figure 1 (b) shows the temporal mean BVs growth rate for the classic and variant L96 models. The variant L96 model has an overall higher growth rate than the classic L96 model (Figure 1 (b)), which agrees with the statement in Lorenz and Emanuel (1998). Moreover, the perturbations tend to grow on the land-sea interface (Figure 1 (c)) and propagate eastward with the group velocity

(Figure 1 (d)). In summary, we expect the variant L96 model to offer more complicated dynamics than the L96 model, and its more rapid error growth would let the improvements made by DA be more quickly lost.



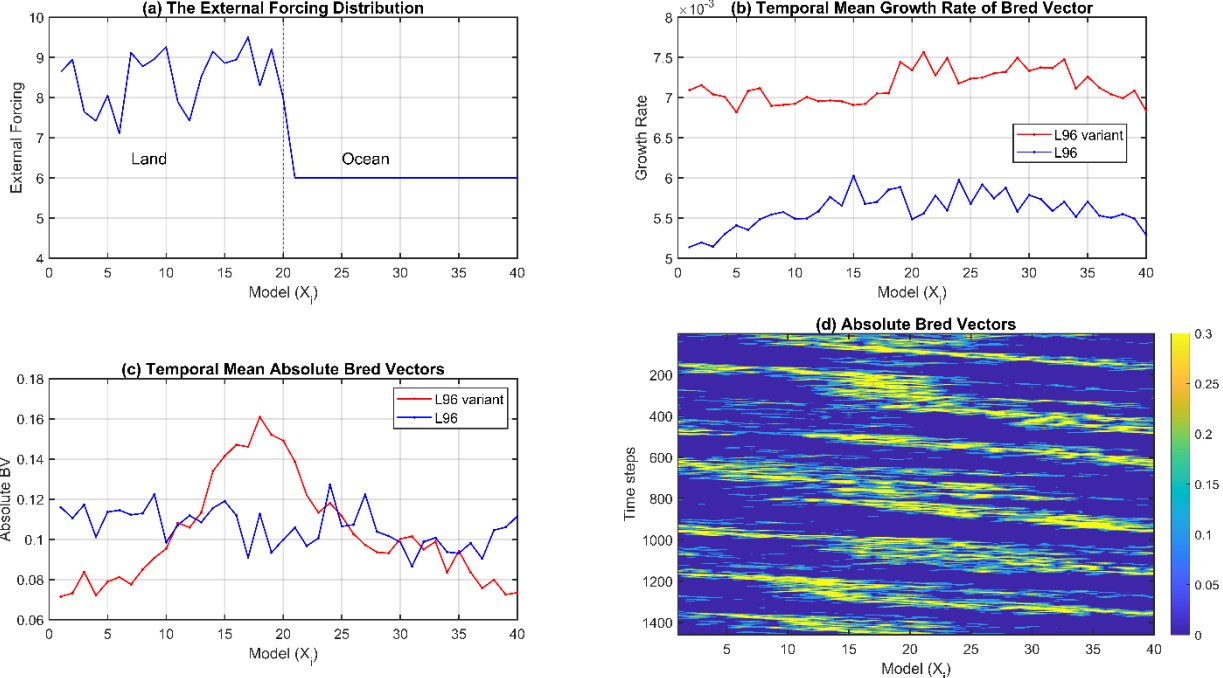

**Figure 1**. (a) The external forcing ($F + f_i$) used in the variant L96 model. The temporal mean of (b) the growth rate and (c) absolute bred vectors. (d) The time evolution of the absolute bred vectors for the variant L96 models. The breeding rescale cycle is 4 steps ($n = 4, \Delta t = 0.0125$), which equals our DA window length. The breeding rescale amplitude is 1.0.

### 3.2 Localization Methods

In this study, we investigated four types of localization strategies:

- **GDL**: Distance-dependent localization introduced in Section 2.3. The localization length used for each experiment is experimentally tuned for a minimum temporal mean analysis RMSE. The cutoff radius is set to be 3.65 times the localization length.

- **YK18**: Correlation-dependent localization, in which the weighting function is derived from the correlation cutoff method (Yoshida and Kalnay, 2018) introduced in Section 2.3.

- **Hybrid**: a hybrid application of GDL and YK18, in which the localization weighting is equal to $\alpha\,GDL + (1 - \alpha)YK18$. The combination ratio $\alpha$ is 0.5 for our experiment. This method was only tested for the classic L96 model experiment.



- **Hybrid II:** Combination use of GDL and YK18, in which YK18 is employed for the first 80 DA cycles for shortening the spin-up, and GDL is subsequently applied for the rest of DA cycles. This method is used only for the variant L96 model experiment.

For YK18, an independent offline run with sequential LETKF DA cycling is conducted for acquiring the prior error correlation before running the DA experiments. The running period for the offline run is three years with a 6-hr analysis window. The first four months are assumed to be the spin-up period and were removed. We used 50 ensembles for the offline run without any localization. Note that the offline runs were performed respectively for the classic and variant L96 models. The multiplicative covariance inflation is applied and optimally tuned for each experiment.

Theoretically, the optimal localization length for GDL is directly proportional to the ensemble size, and there must exist an optimal combination of the localization length and the inflation factor (Hamill et al., 2001). This optimal combination is experimentally defined based on the minimum averaged analysis error for every experiment.

## 3.3 Truth and Observations

The truth was obtained from the model free-run, and the observations were generated by adding random Gaussian errors with a variance of 1.0 onto the truth state every 6 hours. The initial ensembles are obtained from the perturbed model states and integrated for 75 days until the ensemble trajectories converge to the model attractor. The total experiment period is one year.

The analysis result is evaluated by the root-mean-square error (RMSE) with the truth state. For each variable, the RMSE
can be represented as:

$$\text{RMSE} = \sqrt{\frac{1}{M}\sum_{i=1}^{M}(\overline{x_i^a} - x_i^e)^2}\,, \tag{9}$$

where M is the number of model grids, which equals 40 for the L96 model. The $\overline{x_i^a}$ and $x_i^e$ are the analysis ensemble mean and the verified state, respectively.

## 4 Results

### 4.1 Fundamental characteristics of the YK18 function

The squared error correlation estimated from the independent background ensembles is the core of the YK18 localization function. Here, we discussed (1) how different factors (ensemble and observation) in the offline run impact the corresponding error correlation estimation (Eq (6)), and (2) what the main differences in the localization functions (e.g., GDL and YK18) are?





First, we examined the temporal mean squared correlation (Eq (6)) estimated by different observation and ensemble sizes of the offline runs. Trials with observation sizes of 40, 20, and 13 (representing uniform coverages of 100%, 50%, and 30%, respectively) are carried out on the classic L96 model with 40 ensembles. We found that the squared correlation estimation (Eq (6)) is not very sensitive to the observation size changes (Figure 2 (a)), as long as the analysis of the offline run is well constrained. Moreover, the minor differences in the estimated squared error correlation (Figure 2 (a)) would be ultimately
smoothed out by the cutoff function (Eq (7)) in practice. Therefore, the final localization weights derived from the offline runs with different observation sizes will be almost identical.

In contrast, the impact of the ensemble size on the error correlation estimation is more pronounced (Figure 2 (b)). As expected, too few ensembles would induce spurious error correlations, especially in distant regions, and consequently degrades the error correlation estimation. That implies that having sufficient ensembles for the offline run is essential for
generating a reliable error correlation when applying the YK18 method.

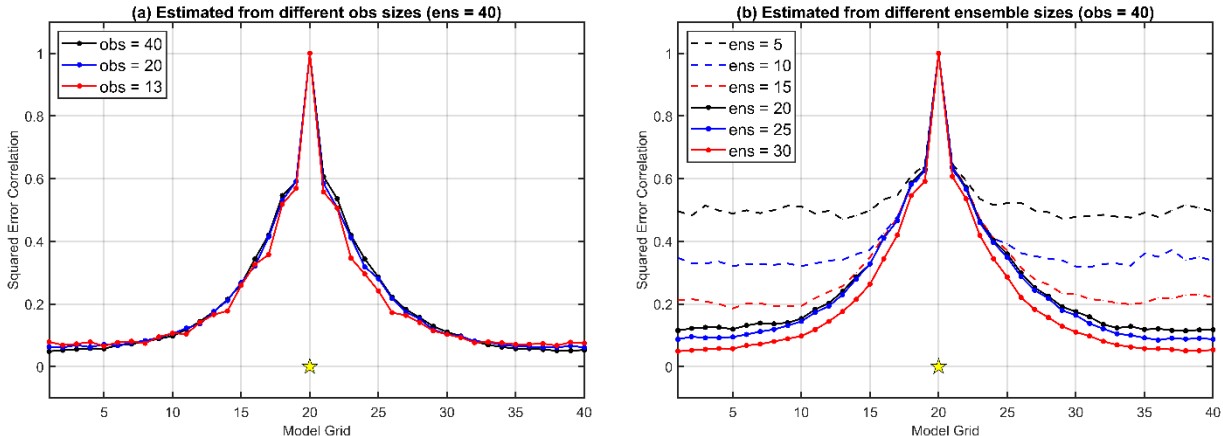

**Figure 2.** The temporal-mean squared error correlations estimated from different (a) observation amounts and (b) ensemble sizes. The yellow star represents the correlated observation location.

The localization functions of GDL and YK18 applied for our DA experiments are shown in Figure 3. In general, the shape of the GDL localization function completely depends on the chosen localization length. The optimal localization length is associated with multiple factors like ensemble size, observation distributions, and model dynamics. For example, when the ensemble size shrinks, the optimal localization length would correspondingly decrease so that a stronger suppression effect can be performed on those spurious correlations in the distant regions. In contrast, the YK18 localization
function, once it is defined, is independent of the ensemble size changes. Unlike GDL, which provides a fixed function for every observation, YK18 offers customized localization functions for each observation based on their prior error correlations (Figure 3 (b) and (c)). Notably, YK18 presents an asymmetric and tighter localization function (Figure 3) than GDL. As shown in Section 4.2, the shape of the YK18 localization function is closer to the actual error correlation and would


contribute to a faster spin-up for the L96 model. We expect the prior correlation information would let YK18 have a more
precise use of the observations, which may partially compensate for the influence of smaller observation impacts given by
the relatively tighter localization function and provide similar performance as GDL after the system convergence.

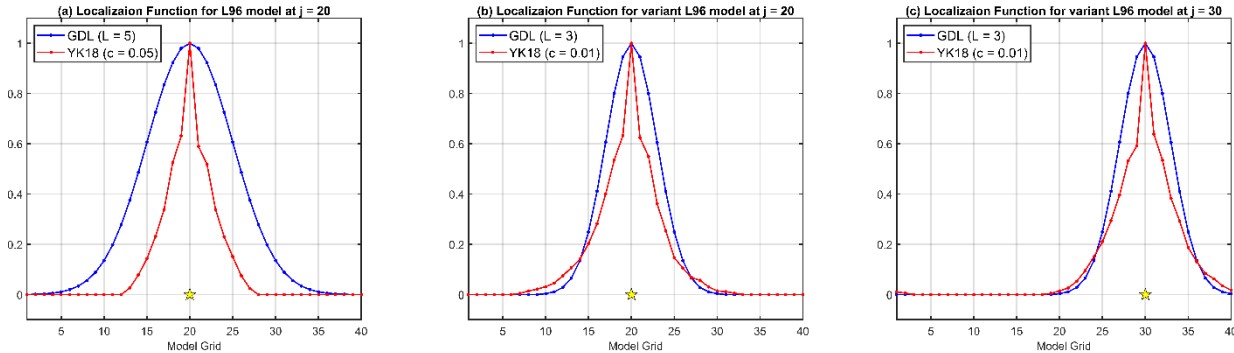

**Figure 3.** The localization functions of GDL (blue) and YK18 (red) for the (a) classic L96 model and (b)(c) the variant L96
model but for the different observation sites. The yellow stars represent the corresponding observation sites. The results
presented here are for the case of 10 ensembles and 40 observations.

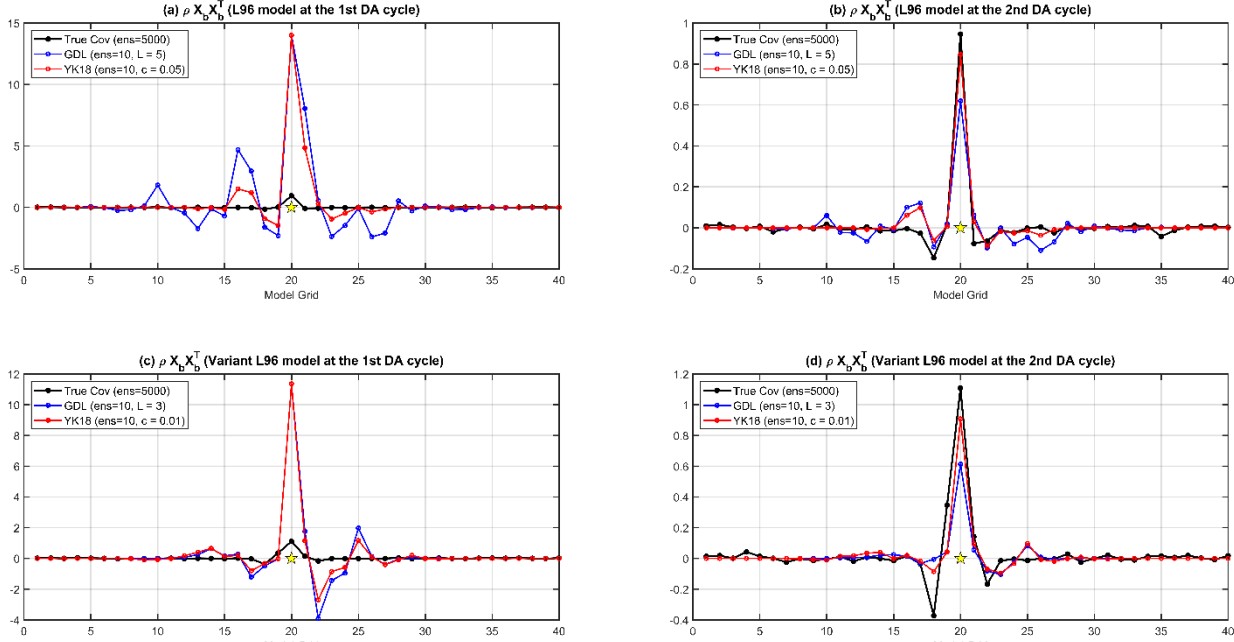





**Figure 4**. The true (black) and localized background error covariances ($\rho\mathbf{X}_b\mathbf{X}_b^T$) of GDL (blue) and YK18 (red) for the L96
model at (a) the first and (b) the second DA cycles, and for the variant L96 model at (c) the first and (d) the second DA
cycles. The localization functions and configurations are the same as in Figure 3.

### 4.2 Scenario I: classic L96 model

In this section, the classic L96 model was utilized to investigate the impacts of GDL, YK18, and Hybrid. The total
experiment period is one year (after the first 60 cycles of spin-up) with a DA window of 6 hours. The tested ensemble sizes
are 8 and 10. Observations are uniformly distributed with a total number of 20 and 40.

Figure 5 shows the analysis RMSE of GDL, YK18, and Hybrid. Table 1 shows the 1-yr mean analysis RMSE without the
spin-up period. In general, the long-term averaged performance of the three localizations is very similar (Table 1), while
GDL is slightly better, and Hybrid is between GDL and YK18. However, YK18 presented significantly lower RMSE than
GDL during the spin-up period (Figure 5), particularly when the ensemble size and observations were reduced (Figure 5 (d)).
This result suggests that YK18, surprisingly, can shorten the spin-up for the DA system and perform a comparable analysis
as GDL.

The capability of YK18 in accelerating the spin-up mainly comes from its more precise interpretation of the error
correlations derived from the independent (or past) ensembles. Figure 4 shows the localized background error covariance
($\rho\mathbf{X}_b\mathbf{X}_b^T$) of GDL (blue line) and YK18 (red line) at the first (Figure 4 (a)) and the second (Figure 4 (b)) DA cycles. The true
covariance (black line) was obtained by perturbing the truth state and evolving through the corresponding DA window (6
hours) with a large ensemble size of 5000, which can be seen as an optimal estimation without sampling errors. At the first
DA cycle, where GDL and YK18 were initialized with the same ensembles, it is apparent that the localized error covariance
of YK18 is significantly closer to the true covariance, particularly for the adjacent grids (e.g., the covariance part) (Figure 4
(a)). With a better estimate of the background error covariance, YK18 performed a superior analysis at the initial cycle and
subsequently improved the background error estimation for the next cycle (Figure 4 (b)). Moreover, this advantage of YK18
is also present in the variant L96 model (Figure 4 (c)(d)). That is, with a prior knowledge of the error correlations, YK18 can
optimize the use of observations, inducing more "on-point" corrections for the analysis and reducing the required number of
cycles for the DA system's spin-up.





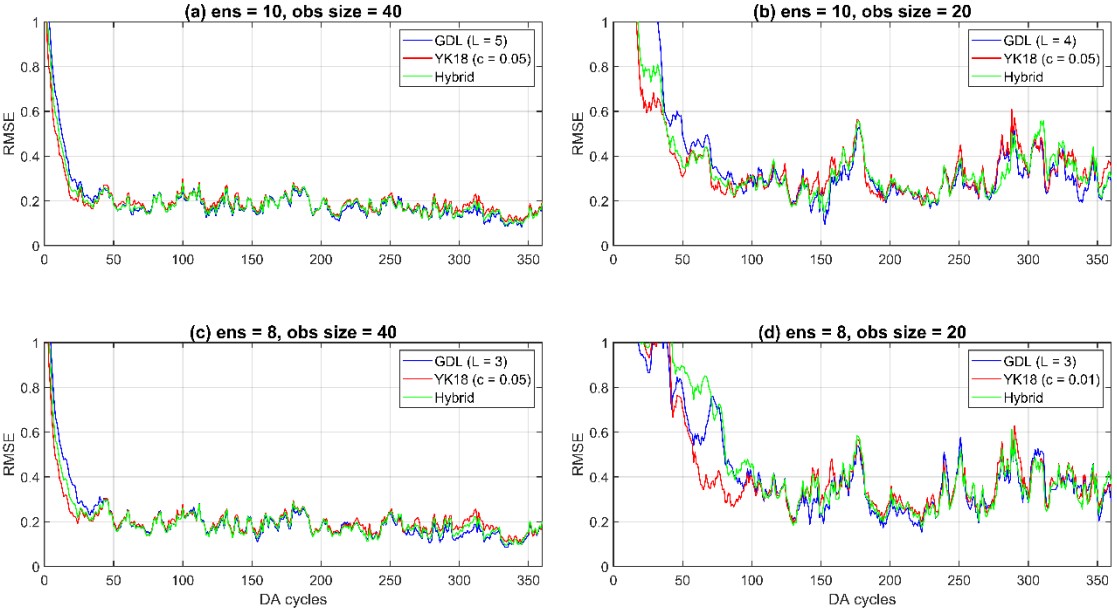

**Figure 5.** The time series of the analysis RMSE for GDL (blue line), YK18 (red line), and Hybrid (green line) for the cases of 10 ensembles with (a) 40 and (b) 20 observations; and cases of 8 ensembles with (c) 40 and (d) 20 observations.

**Table 1.** The long-term mean analysis RMSE for the classic L96 model

|  | Observation = 40 | | Observation = 20 | |
| --- | --- | --- | --- | --- |
|  | Ensemble = 8 | Ensemble = 10 | Ensemble = 8 | Ensemble = 10 |
| GDL | 0.178 | 0.175 | 0.292 | 0.245 |
| YK18 | 0.197 | 0.193 | 0.309 | 0.278 |
| Hybrid | 0.186 | 0.182 | 0.298 | 0.256 |

**4.3 Scenario II: the variant L96 model**

Considering the classic L96 model is favorable for GDL due to its simple model dynamics (Table 1), the variant L96 model that offers a more complicated model dynamic was employed here. We used ten ensembles and tested with different observation sizes of 40, 30, and 20. The 20 and 40 observations are uniformly distributed. The 30 observations are distributed densely on the land (20 observations) region and coarsely on the ocean area (10 observations). Here, three localization methods were tested: GDL, YK18, and Hybrid II. Hybrid II is a mixed-use of GDL and YK18, where it uses




YK18 for the first 80 DA cycles for accelerating the spin-up, then GDL for the rest of the cycles. Note that the optimal localization length is respectively tuned for each method, so the localization length used in GDL and Hybrid II may be different.

285    Figure 6 shows the analysis RMSE of the three methods on the variant L96 model. Note that Hybrid II is identical to YK18 for the initial 80 DA cycles, so the green overlaps with the red line in Figure 6. As expected, GDL requires a significantly longer spin-up for the more complex model, especially when fewer observations were served (Figure 6 (b) and (c)). The YK18, again, showed impressively efficiency in accelerating the spin-up, particularly with fewer observations, and generated a better analysis than GDL at the early stage (Figure 6). Moreover, this advantage of YK18 became more pronounced with a more complicated model and fewer observations.

290

**Table 2.** The long-term mean analysis RMSE for the variant L96 model (10 ensembles)

|          | obs = 40 | obs = 30 | obs = 20 |
|----------|----------|----------|----------|
| GDL      | 0.147    | 0.200    | 0.233    |
| YK18     | 0.160    | 0.203    | 0.258    |
| Hybrid II | 0.128    | 0.160    | 0.201    |

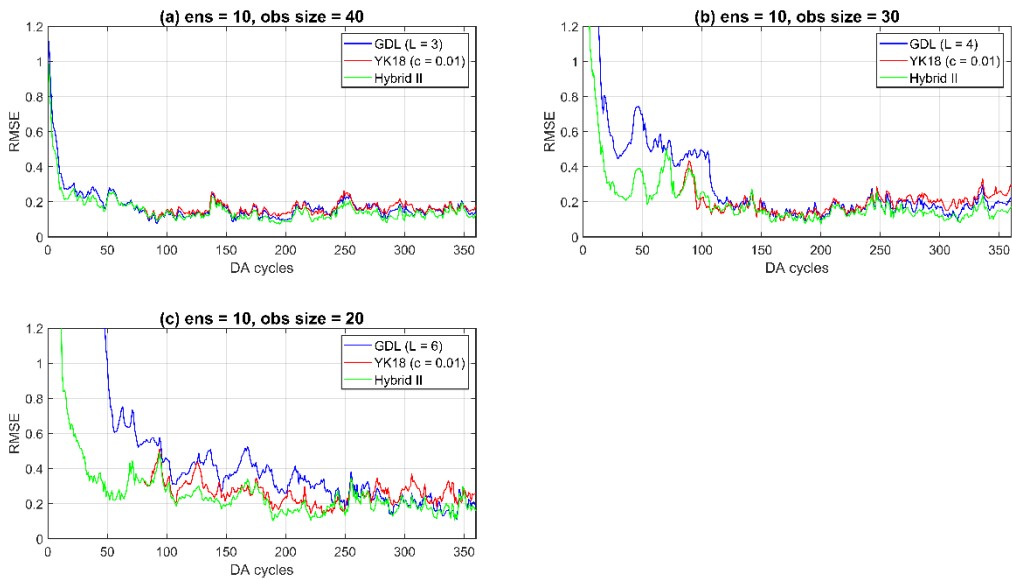

**Figure 6.** The analysis RMSE of the GDL (blue), YK18(red), and Hybrid II (green) with observations of (a) 40, (b) 30, and
295   (c) 20 for the variant 96 model experiment.





Table 2 is the 1-yr average of the analysis RMSE after the first 100 spin-up cycles. After the system's spun-up, the averaged analysis RMSEs of all methods are similar, while Hybrid II is slightly better than the other two methods (Table 2). It is surprising that Hybrid II, the mixed-use of YK18 and GDL, is superior to solely using YK18 or GDL. Hybrid II inherits the benefit of YK18 of accelerating spin-up and outperforms GDL after the system convergence, presenting the best performance among all methods. That is, possibly, because Hybrid II has a longer optimal localization length than GDL, allowing it to acquire more observation information during the assimilation and provide a more accurate analysis.

Finally, it is important to highlight that YK18 is an exceptionally efficient localization method. In practice, the use of GDL requires multiple preceding trials to define an optimal length for the experiments of interest, which may consume considerable computational resources and time. In contrast, YK18 only needs one offline run to determine the error correlations, whereas it performs a comparable analysis as GDL, even with a faster spin-up.

## 5. Summary and Discussion

This study explored the feasibility of using the correlation cutoff method (YK18, Yoshida and Kalnay 2018; Yoshida 2019) as a spatial localization and compared the accuracy of the two types of localization, the correlation-dependent (YK18) and distance-dependent (GDL), preliminarily on the Lorenz (1996) model with the LETKF. We also proposed and explored the potential of the two types of hybrid localization applications (Hybrid and Hybrid II). Our results showed that YK18 performs a similar analysis as GDL but with a significantly shorter spin-up, especially when fewer ensembles and observations are presented. YK18 can accelerate the spin-up by optimizing the use of observations with its prior knowledge of the actual error correlations, effectively reducing the required number of cycles toward the analysis convergence. In our experiments with the variant L96 model, we demonstrated that these advantages of YK18 would become even more pronounced under a more complicated dynamic.

It is worth highlighting that YK18 is more efficient and economical localization than GDL. Traditionally, the use of GDL requires multiple trial-and-errors to define the optimal localization length for the experiments of interest. In contrast, YK18 only needs one offline run to obtain the prior error correlations, whereas it provides a comparable analysis as GDL even with a faster spin-up. For operational or research centers that have plentiful archives of historical ensemble datasets, it is possible to directly obtain the required prior error correlation for YK18 from the past data without executing the offline runs in advance.

We found that Hybrid II, a combination of YK18 and GDL, generated a more accurate analysis than that solely using GDL or YK18. Hybrid II has the same advantages as YK18 in accelerating the spin-up and a larger optimal localization length than GDL. These features allow Hybrid II to spin up quicker, obtain more observation information after the system convergence, and generate a slightly better analysis than GDL and YK18. Since the analysis unbalances would be relaxed by





a larger localization length (Lorenc 2003; Greybush et al., 2011), we expect Hybrid II would deliver a more balanced analysis than GDL with a multivariate model. Further investigation of this advantage will be part of our future works.

Finally, we would like to emphasize that the L96 model used in this study is highly advantageous to GDL because of its
univariate and simple dynamic without teleconnection features. So, the two known problems in GDL, unbalanced analysis and losing long-range signals, would not appear here to degrade its performance. Despite that, this model is still an excellent testbed for preliminary DA studies because it offers a simple and ideal environment for first exploring the fundamental characteristics of new methods. With that in mind, it is encouraging that YK18 performed a comparable analysis to GDL (even with shorter spin-up) under such an environment that is particularly advantageous to GDL. We believe YK18 has a
great potential in generating a relatively accurate and balanced analysis than GDL in a more sophisticated, multivariate model. More studies with a multivariate and more realistic model would be required and will be conducted as our future works.

**Code availability**

The codes for the methods can be provided by the corresponding authors upon request.

**Author contributions**

CCC and EK designed the concept of the study. CCC developed the code and performed experiments. EK provided the idea of the L96 variant model and guidance for all the DA experiments. CCC wrote the manuscript, and EK reviewed and edited
it.

**Competing interests**

The authors declare that they have no conflict of interest.

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
