# Peer review of "Applying prior correlations for ensemble-based spatial localization"

_Nonlinear Processes in Geophysics, 2022_

## Author Response (AR1)

Dear editors and reviewers,

Thank you for all the comments on our manuscript entitled "Applying prior correlations for ensemble-based spatial localization". We greatly appreciate the interests that the editors and the reviewers have taken in our manuscript and the constructive comments they have given.

All the comments and suggestions were very helpful for revising and improving our paper. We have studied these comments carefully and have made corresponding corrections that we hope will meet with your approval. More specifically, we have made the following significant changes in this revision:

1) All the results have been updated with a more practical and convincing configuration. We have re-conducted the YK18 and related methods by offline runs with the same ensemble size as the DA experiments. Contrary to the affirmations of reviewer#1, our new experiments show that the YK18 was not found to be flawed.

2) We have greatly improved the clarity and flow of our manuscript, including adding more references, more explicit interpretations of the results and settings, and corrections on some minor writing errors.

3) Discussions related to the real applications of our method (i.e., location varying observations, implementing a large model) have been added to the manuscript.

A point-by-point response to the reviewers' comments is included in the Reply (the reviewer's comments are in italics). Changes in the revised manuscript are tracked and highlighted.

Thank you again for your consideration of our revised manuscript. If you have further queries, please do not hesitate to contact us.

Sincerely,

Chu-Chun Chang

Corresponding Author,

University of Maryland, College Park, U.S.A

** **reviewer's comments in black and italics   Author's comments in red**

*Reviewer 1*

*This manuscript applies a newly developed spatial localization method (YK18) for the ensemble-based data assimilation using Lorenz (1996) model. This correlation cutoff method is developed in their previous work as a variable localization strategy for coupled systems. They claimed that it can be further utilized as a spatial localization method. They performed twin experiments with Lorenz 1996 model, and compared the YK18 method with the conventional spatial localization method. Overall, the work is useful and the manuscript is well written. However, some details of the method should be better explained. And I still have questions about the foundation of the method. The authors should answer my questions and make some revisions before it could be accepted for publication. Please see my questions and comments below.*

**Reply:**

We sincerely thank the reviewer for the questions that help us improve the clarity of our manuscripts. The reviewer provided comments about the feasibility and the foundation of our method, especially for the extensive use of YK18 from the L96 model to sophisticated models.

We would like first to emphasize that it is common to use the L96 model as a first step to examine the feasibility of new approaches, such as another amazing localization work done by Anderson (2007). The applications to large models, including the multivariate effects and practical strategies, will be explored in our future studies.

In this revised manuscript, we have incorporated the reviewer's comments and updated our experiments with an offline run with the same ensemble size as the experiment. **The new results proved that the YK18 could be derived from past data or offline runs with limited ensembles, by which the reviewer's main concerns are likely to be resolved.** The detailed answers for each question are attached below.

If our answer and interpretation are not clear enough to the reviewer, we would be pleased to have a meeting with the reviewer to discuss it in more detail.

*1.      In section 2.3, the authors imply that "The prior square error correlations are collected from a preceding offline run." According to section 3.2 (line 185), I realize the offline run is a DA experiment using a relatively large ensemble (50 in this case) without localization for a long period. Of course it works with toy models such as the Lorenz model in this work and that in YK18. However, for more complicated models (such as GCMs), it is meaningless to perform DA experiment without localization, and it is impossible to use an ensemble large enough to get rid of localization. That would weaken the argument about the usefulness of the method.*

**Reply:**

We appreciate and agree with your comments. In this revised manuscript, we have updated all experiments with YK18 using only ten ensembles (the same as DA experiments) and localization for the offline runs. We have updated our results in Sections 4.2 and 4.3. The results proved that it is possible to use limited ensembles or historical ensemble forecasts for YK18 with a proper tunning of Eq (6). Thus, it is totally possible to apply YK18 on large models.

*2.    Equation (6) implies that the temporal mean of the squared correlation over all analysis steps is computed to serve as "prior error correlation" to estimate the localization function. I have some questions about that:*

*2.1 Why do you compute the temporal mean over all analysis steps, instead of all steps including forecast and analysis?*

**Reply:**

There are two reasons why we choose the ensemble background (forecast) at each analysis step for estimating the error correlation:

- **The availability of data**

    In practice, the model states (i.e., forecasts and analysis) and observations are mainly available at the regular analysis times (i.e., 00Z, 06Z, 12Z, 18Z) on a DA routine basis. Therefore, it is reasonable to assume that the data we can use are only available at each analysis time.

- **The potential issue of the error correlation in the analysis**

    We avoid computing the error correlations from the analysis because the relationship between model variables and the observation in the analysis could differ from the relations in the physical system. Also, the assumptions (i.e., linear regression, localization) used in DA could induce artificial sampling errors in the analysis (Anderson 2007), thereby affecting the error correlation estimates. For example, the use of the traditional localization could introduce an unbalance of the analysis state (i.e., ageostrophic wind) (Kepret 2009; Greybush et al., 2011) if no post-processing (i.e., IAU) proceeds for the analysis. Usually, the imbalance can be eliminated by the model itself after a few integration steps. Therefore, obtaining the prior error correlations from the forecasts at the end of the DA window (= the background at each DA time) and not including the analysis would be an appropriate configuration.

*Considering the analyses are from the data assimilation without localization, does that indicate the ensemble size is large enough such that the true correlation can be recovered without localization? This is still impossible for large models.*

**Reply:**

Yes, we initially applied large ensembles for the offline run without localization to better capture the true correlations. As we mentioned earlier, we have updated all the experiments with offline

runs with limited ensembles in the revised manuscript. Therefore, it would be totally possible for our method to be applied to large models, and your primary concern is well resolved now.

*2.2 You use the temporal mean of the squared correlation. So, does the period of the offline run have an impact on the correlations?*

**Reply:**

Once the temporal mean of the squared correlation is *converged* to the climatology, the result from different periods of offline runs would be the same. The required period of the offline run (i.e., number of samples) depends on the configurations (i.e., ensemble size) and model complexity.

In this revised manuscript, we have added a new Figure 2 (b) discussing the required periods of runs for different models and ensemble sizes.

[Figure]

*2.3 Is the assimilation process necessary in the offline run?*

**Reply:**

Yes, the DA process is necessary to constrain the system, preventing the state from jumping to another phase or having diverged ensembles, especially for a very chaotic system such as the atmosphere. Besides, the offline run was designed to mimic the DA cycle adopted by operational models, where the analysis cycle is performed every 6 hours.

*3. I wonder, is it possible to compute the correlation using an EnOI-like idea? i.e., running a single model and computing the correlation using members at different time steps. This seems much more practical for real applications. You have already used the temporal mean in the current method anyway.*

**Reply:**

Thank you for this interesting suggestion. After careful research, we found that the feasibility of EnOI-like method depends on the characteristics of the dynamic system. Below we will interpret this argument in detail.

First, assume we have two methods to obtain the prior error correlation:

- **YK18**: correlation derived from ensembles at each DA time, then do the temporal mean over the running period.
- **EnOI-like**: correlation derived from sampling the single model states (ex. ens-mean analysis/forecast) at different time steps along the running period.

Assuming observations are located at the model grid (so, h(x)=x), then the correlation formulations for YK18 and EnOI-like are:

$$Corr_{ij}(YK18) = \frac{1}{T}\sum_{t=1}^{T}\left(\frac{\boxed{\sum_{k=1}^{K}[x_{k_i}(t) - \overline{x_i(t)}]\,[x_{k_j}(t) - \overline{x_j(t)}]}}{\sqrt{\sum_{k=1}^{K}[x_{k_i}(t) - \overline{x_i(t)}]^2}\,\sqrt{\sum_{k=1}^{K}[x_{k_j}(t) - \overline{x_j(t)}]^2}}\right)$$

$$Corr_{ij}(EnOI-like) = \frac{\boxed{\sum_{t=1}^{T}[\bar{x}_i(t) - \bar{C}_i][\bar{x}_j(t) - \bar{C}_j]}}{\sqrt{\sum_{t=1}^{T}[\bar{x}_i(t) - \bar{C}_i]^2}\,\sqrt{\sum_{t=1}^{T}[\bar{x}_j(t) - \bar{C}_j]^2}}$$

Here, we follow the same symbol as Eqs (5) and (6) in our paper, $C_i$ is the temporal mean of the selected samples of $\underline{x}$ (equals to the climate mean of the samples).

One can notice that their covariance parts (i.e., part A) actually reflect different physical meanings: YK18 is the temporal-mean uncertainty, representing the background error in EnKFs, and the EnOI-like is the anomaly to the climatology. The figure below is a schematic of these two differences:

[Figure]

For a dynamic system like the ocean that is overall stable (less chaotic) and its variability is generally proportional to the anomaly, it is possible to use the EnOI-like method to represent the error statistics of the system. However, for a very chaotic system such as the atmosphere, the errors could grow very quickly while its anomaly remains small (like the schematic figure below). For such a case, using an EnOI-like method may lose the signals of short-term perturbations and thus underestimate the error correlations of the system.

[Figure]

To examine the above hypothesis, we conducted a 2-year experiment on the L96 model with YK18 and EnOI-like methods. The figure below is the prior error correlation estimated by the EnOI-like method (green line) and YK18 with offline runs (red) and past runs (blue). The offline run with a large ensemble (red) can be seen as the "true" correlation. **The result confirmed our expectations that the EnOI-like method may underestimate the error correlations for a very chaotic system such as the L96 model.**

[Figure]

*4. About Figure 2a. I cannot see any connection between error correlation and observation size from equations (5) and (6). But the connection between error correlation and ensemble size is very clear from equation (5). Do you use this figure to explain the parameters in the offline run?*

**Reply:**

Yes, the aim of Figure 2 is to show how different parameters (observation and ensemble size) used in the offline run could affect the prior error correlation estimations.

As we stated in [Line 212 - 213], Figure 2(a) confirms that "the prior correlation estimation (Eq (6)) is not very sensitive to the observation size changes, as long as the analysis of the offline run is well constrained ". This conclusion is very important for the YK18 application with new instruments. Assuming we have ensembles from past runs that only assimilated 20 observations. Then, we want to increase the observations to 40 for the new experiment. From Figure (2a), we know that it is possible to directly obtain the prior error correlation from the 20 obs past runs (because eq(5) and (6) can be computed directly from past ensembles as long as we know the observation location j). This characteristic, in other words, provides evidence for YK18 to use past

data to estimate the localization function for newly added observations, which is a significant feature for the applicability of YK18 in modern DA.

**We have modified sentences in Section 4.1 to highlight the importance of the result of Figure 2 (a) in this revised manuscript.**

5.      *The comparison results in figure 5 and figure 6 are not impressive. Though the authors declaim that YK18 can accelerate the spin-up. However, the parameters in GDL and YK18 may be not optimal, so the conclusions are not very persuasive.*

**Reply:**

We appreciate your comments and would like to make further clarifications here.

First, our conclusions are drawn from the optimal configurations for all methods. As we mentioned in the manuscript [Line 190], all the parameters used in GDL and YK18 were optimally tuned for a minimum mean analysis RMSE. So, the parameters we applied in the paper were optimal for GDL and YK18. We conducted experiments with careful assumptions and configurations, so the results we presented are scientifically meaningful and compelling.

In this revised manuscript, we have added Tables listing the parameters used for each experiment. Also, we have improved the clarity of the statement of our configurations in Section 3.2.

*Detailed comments*

- *Line 175 "GDL: Distance-dependent localization introduced in Section 2.3." I think it is section 2.2.*

**Reply:**

Thank you for pointing this mistake out. We have corrected it in the revised manuscript.

- *For equations (1) - (3), there are linear observation operator H, for equation (5), it is a potentially nonlinear operator h(x)*

**Reply:**

Thank you for the comment about the symbol. The capital H in Eqs (1)-(3) represents the matrix with elements of linear operator h. As we specified in [line 115], h(x) is an element-wised interpolation:

"$hj(xk(t))$ is the linear interpolation to the background state $xk(t)$ from the analysis grid to the *jth* observation location".

- *Line 99 "Equation (4) is a smooth and static Gaussian-like function that offers the same localization effect as the GC99 when applied to LETKF." It is inaccurate, because GC99 uses a compact-support function, and it cutoff at some distance, but Eq. (4) does not cutoff.*

**Reply:**

Thank you for the question. Please note that we already covered the compact-support part in the same paragraph [line 105] in our manuscript:

*"When the compact support is presented with the localization function, the observations located beyond a certain distance (in this study is 3.65 times L) from the analysis grid would be discarded by assuming $\rho ij$=0. "*

- *R1: Line 91 and line 102, whether the R localization multiplies the elements of R inverse or R itself? Please clarify that.*

**Reply:**

Thank you for pointing this out. It should be R itself. We have corrected this part for [Line 91] in the revised manuscript.

*Reviewer 2*

**Reviewer's comments in black and italics    Author's response in red**

We are very grateful to Reviewer 2 for understanding our method well and for providing many constructive comments, including comparing Anderson's work and applications to special grids and location-varying observations. These comments are very insightful and helpful in improving our manuscript.

In this revised manuscript, we not only improved the clarification of the method and settings, and also incorporated the reviewer's comments in the discussion. The point-by-point responses to the questions are attached below.

*Reviewer 2*

*The authors propose a localization scheme for prior correlations and compare it with the traditional localization scheme based on distance dependence. This localization scheme is of interest for the implementation of ensemble data assimilation methods. The manuscript is quite well written and meets the submission requirements of the journal NPG. Nevertheless, the manuscript has the following issues that need further clarification and improvement.*

*Specific comments:*

1. *This new localization scheme for YK18 relies heavily on the statistical formulation of equation (5), so is there any similarity between this formulation and Anderson's work, and what are their similarities and differences? Please elaborate explicitly.*

**Reply:**

Thank you for the valuable comment. We have added Anderson's work in the Introduction section in this revised manuscript.

Both Anderson's work (hereafter AL13) and our method (YK18) are proposed to obtain a static localization function from posterior ensembles. Both methods deliver a flow-dependent localization and show comparable accuracy to the traditional localization method with the L96 model. However, there are several differences between our method and AL13.

AL13 tends to find the localization weight that performs a minimum analysis ensemble-mean RMSE. This work is achieved by minimizing the cost function for a group of ensembles (subset ensembles). The minimization is carried out with an OSSE with "truth" values. Moreover, iterations for solving the equation would be required. In addition, AL13 does not restrict its

solution from 0 to 1, so the localization weight obtained from AL13 could be a negative value or exceed one.

In contrast, YK18 finds a localization by the prior error correlations from the ensembles generated by an offline run or from past data. It does not need a truth value nor runs iteratively like AL13, while it needs an additional cutoff function to filter out small perturbations in the prior error correlations. Besides, the localization weight obtained from YK18 is restricted to an interval between 0 and 1.

*Does the statistical result of Equation (5) depend on the number of samples? If so, how much does this sample dependence affect the final results?*

**Reply:**

Once the solution of Eq (5) is converged to the climatology, the final result would be almost the same. The required number of samples depends on the configurations (i.e., ensemble size) and model complexity.

In our revised manuscript, we have added a new Figure 2 (b) showing the convergence time for different models and ensemble sizes. A related discussion has been incorporated into Section 4.1.

*Equation (5) counts the correlation coefficients between the model grid points and the observed points, but we know that the observed variables are hardly fixed in their positions at different moments. This situation is especially prominent when assimilating satellite data in NWP. Since the position of the observed data is difficult to be fixed, the observation operator H is actually difficult to be fixed as well. Then how should the correlation coefficients between the model grid points and the observed points, which are calculated by Eq. (5), be applied to other moments?*

**Reply:**

Thank you for this great question! We have incorporated the related discussion in Section 5 as future work.

For the observations that their position varies with time (i.e., satellite data), a possible solution is the application of machine learning. Yoshida (2019) showed that neural networks could be used to estimate the background error correlations and deliver the correlation function for YK18. This work can be further applied to varying observations for estimating their prior error correlations and will be our future works.

*Similarly, the model in the validation experiment given so far is very simple, with only one variable. For a true NWP model, there are perhaps multiple model state variables such as U, V,*

*P, T, Q, on the same model grid point. And due to the use of different grid schemes, these variables may not appear at the same location of the grid. So how to use Eq. (5) for statistics in this case and apply it to the real situation?*

**Reply:**

Thank you for this valuable and important question. For a multivariate system, Eq. (5) can be directly calculated for different pairs of model variables. For example, Eq (5) can be represented as the correlation between U ($x^u$) and V ($x^v$) as:

$$corr_{i_j}(t) = \frac{\sum_{k=1}^{K}\left[x_{k_i}^u(t) - \overline{x_i^u(t)}\right]\left[h_j\left(x_k^v(t)\right) - \overline{h_j\left(x_k^v(t)\right)}\right]}{\sqrt{\sum_{k=1}^{K}\left[x_{k_i}^u(t) - \overline{x_i^u(t)}\right]^2}\sqrt{\sum_{k=1}^{K}\left[h_j\left(x_k^v(t)\right) - \overline{h_j\left(x_k^v(t)\right)}\right]^2}}$$

For special model grids such as the staggered grid in WRF or cell grids in MPAS, there are several strategies (i.e., convert to Cartesian coordinate, direct interpolation…etc.) proposed to deal with the grid structure problem in DA (Ha et al., 2017; Pattanayak and Mohanty,2018). Considering the fundamental statistics of YK18 (Yoshida and Kalnay, 2018), we would suggest calculating Eq (5) with the analysis grid and observation operator that are consistent with the term of **HXX**$^T$ in the corresponding DA system.

How to properly apply localization, for either GDL or YK18, on special model grids is an important and challenging topic that needs more investigation. This advanced topic is beyond the scope of the current paper, and we will include it in our future works.

*The authors elaborate that one of the advantages of YK18 is that it is more computationally efficient. However, it can be seen from their analysis that in fact YK18 should essentially provide some new calculations of localization correlation matrices as well, so why does it make the improvement of computational efficiency?*

**Reply:**

Thank you for this comment. Yes, both methods need to calculate the localization matrix; however, the pre-processing work for YK18 is more efficient than GDL in two ways:

**1) Fewer trials and errors running for localization:**

As we already mentioned in Section 4.3:

"Traditionally, using GDL requires multiple trial-and-error to define the optimal localization length for the experiments of interest. In contrast, YK18 only needs one offline run to obtain the prior error correlations, whereas it provides a comparative analysis as GDL even with an even faster spin-up."

**2) Fewer computational loops between observations and model grid at each DA cycle**

From the programming aspect, the most expensive part of GDL is the do-loop structure and data sorting. Looping over every model point and observation is essential for calculating the distance before each DA, and data sorting is commonly used for efficiently searching the neighbored observations for the analysis grid. When applying a large model and observations in NWP, the computational costs of looping and data sorting would be significant.

In contrast, the most expensive part for YK18 is developing the prior error correlation in the beginning. However, once the prior error correlation map is constructed, one can directly obtain the correlation value from the indices of i and j in the map (Eq (6)), then convert it to the corresponding localization weight. Therefore, the computationally expensive data sorting during DA can be avoided, making YK18 a more efficient method in the long term.

In the revised manuscript, we have improved the writing of this part in Section 4.3. We are grateful for this comment that helped us improve the manuscript.

*As for the "a faster spin-up" proposed in the manuscript, I do not quite understand it. The purpose of our data assimilation is to give a more accurate initial field and then drive the model to forecast. The spin-up seems to be more appropriate in the simulation of climate models.*

**Reply:**

Thank you for pointing this out. We have added sentences to better clarify the word "spin-up" in our revised manuscript [Line 234-236]. The word "spin-up" in the manuscript is the "DA spin-up", not the "model spin-up".

*It seems that Section 2.3 of GDL appearing in Page7 should be Section 2.2.*

**Reply:**

Thank you for pointing out this mistake. We have corrected it in this revised manuscript.

References:

Ha, S., Snyder, C., Skamarock, W. C., Anderson, J., & Collins, N. (2017). Ensemble Kalman filter data assimilation for the Model for Prediction Across Scales (MPAS). Monthly Weather Review, 145(11), 4673-4692.

Pattanayak, S., & Mohanty, U. C. (2018). Development of extended WRF variational data assimilation system (WRFDA) for WRF non-hydrostatic mesoscale model. Journal of Earth System Science, 127(4), 1-24.